# Associations of Circadian Eating Pattern and Diet Quality with Substantial Postpartum Weight Retention

**DOI:** 10.3390/nu11112686

**Published:** 2019-11-06

**Authors:** See Ling Loy, Yin Bun Cheung, Marjorelee T. Colega, Airu Chia, Chad Yixian Han, Keith M. Godfrey, Yap-Seng Chong, Lynette Pei-Chi Shek, Kok Hian Tan, Ngee Lek, Jerry Kok Yen Chan, Mary Foong-Fong Chong, Fabian Yap

**Affiliations:** 1Department of Reproductive Medicine, KK Women’s and Children’s Hospital, 100 Bukit Timah Road, Singapore 229899, Singapore; jerrychan@duke-nus.edu.sg; 2Duke-NUS Medical School, 8 College Road, Singapore 169857, Singapore; tan.kok.hian@singhealth.com.sg (K.H.T.); lek.ngee@singhealth.com.sg (N.L.); 3Singapore Institute for Clinical Sciences, Agency for Science, Technology and Research (A*STAR), 30 Medical Drive, Singapore 117609, Singapore; marjorelee_colega@sics.a-star.edu.sg (M.T.C.); yap_seng_chong@nuhs.edu.sg (Y.-S.C.); lynette_shek@nuhs.edu.sg (L.P.-C.S.); mary_chong@nus.edu.sg (M.F.-F.C.); 4Programme in Health Services & Systems Research and Center for Quantitative Medicine, Duke-NUS Medical School, 8 College Road, Singapore 169857, Singapore; yinbun.cheung@duke-nus.edu.sg; 5Tampere Center for Child Health Research, University of Tampere and Tampere University Hospital, Arvo Ylpönkatu 34 (ARVO B235), 33014 Tampere, Finland; 6Department of Obstetrics & Gynaecology, Yong Loo Lin School of Medicine, National University of Singapore, National University Health System, 1E Kent Ridge Road, Singapore 119228, Singapore; chiaairu@u.nus.edu; 7Department of Nutrition and Dietetics, College of Nursing and Health Science, Flinders University, Sturt Rd, Bedford Park, SA 5042, Australia; chad.han@flinders.edu.au; 8Department of Dietetics, National University Hospital, National University Health System, 5 Lower Kent Ridge Rd, Singapore 119074, Singapore; 9Medical Research Council Lifecourse Epidemiology Unit, University of Southampton, Southampton SO16 6YD, UK; kmg@mrc.soton.ac.uk; 10National Institute for Health Research Southampton Biomedical Research Centre, University of Southampton and University Hospital Southampton National Health Service Foundation Trust, Southampton SO16 6YD, UK; 11Department of Paediatrics, Yong Loo Lin School of Medicine, National University of Singapore, National University Health System, 1E Kent Ridge Road, NUHS Tower Block Level 12, Singapore 119228, Singapore; 12Khoo Teck Puat-National University Children’s Medical Institute, National University Hospital, National University Health System, 5 Lower Kent Ridge Road, Singapore 119074, Singapore; 13Department of Maternal Fetal Medicine, KK Women’s and Children’s Hospital, 100 Bukit Timah Road, Singapore 229899, Singapore; 14Department of Paediatrics, KK Women’s and Children’s Hospital, 100 Bukit Timah Road, Singapore 229899, Singapore; 15Saw Swee Hock School of Public Health, National University of Singapore, 12 Science Drive 2, Singapore 117549, Singapore; 16Lee Kong Chian School of Medicine, Nanyang Technological University, 11 Mandalay Road, Singapore 308232, Singapore

**Keywords:** circadian eating, diet quality, eating episodes, fasting, meal frequency, pregnancy diet, postpartum weight

## Abstract

Besides food quantity and quality, food timing and frequency may contribute to weight regulation. It is unclear if these factors during pregnancy can influence maternal weight retention after childbirth. We thus aimed to examine the associations of maternal circadian eating pattern and diet quality in pregnancy with substantial postpartum weight retention (PPWR) at 18 months in an Asian cohort. We assessed circadian eating pattern and diet quality of 687 women using 24-h dietary recalls at 26–28 weeks’ gestation. We calculated PPWR by subtracting maternal weight in the first trimester from weight at 18-month postpartum and defined substantial PPWR as ≥5 kg weight retention. Multivariable binary logistic regression was performed. Overall, 16% of women had substantial PPWR. After the confounders adjustment, night eating, defined by greater night-time caloric intake (odds ratio 1.95; 95% confidence interval 1.05, 3.62), and lower diet quality, classified by median score of the Healthy Eating Index (1.91; 1.17, 3.10), were independently associated with higher odds of substantial PPWR. No associations with substantial PPWR were observed for night fasting duration and number of eating episodes. In conclusion, alignment of eating time with day–night cycles and diet quality during pregnancy may play a role in PPWR, with possible implications for long-term obesity risk.

## 1. Introduction

Postpartum weight changes are known to affect women’s long-term weight trajectories [1]. Previous studies reported that at least 14–20% of women were >5 kg heavier than their pre-pregnancy weight at 6–18 months postpartum [1,2]. Retaining more weight after the first year postpartum is associated with a higher body mass index (BMI) at 15 years later [3]. Furthermore, childbearing can increase visceral fat independent of increments in overall body fat [4]. Overweight and obesity due to excess postpartum weight retention (PPWR) have a profound effect not only on future pregnancies, but also on the later health of offspring [5] and lifelong maternal health consequences.

Identifying the dietary predictors of PPWR could direct behavioral intervention efforts for weight management. Randomized controlled trials involving diet interventions for postpartum weight reduction have commonly focused on the amount and type of food consumed [6]. Recent evidence suggests that aligning eating and fasting patterns with day–night cycles, otherwise known as the circadian eating pattern, is important for metabolic health [7]. The circadian gene expression in peripheral cells, especially in the liver, is strongly connected to feeding [8], nutrition, and metabolism, which are consequently closely linked with circadian rhythms [7]. Disruption to the circadian rhythms due to misalignment between circadian clocks and eating patterns can therefore disturb homeostasis and impact on metabolic dysfunction [7].

Independent of total daily caloric intake, studies have shown that higher caloric consumption in the evening/night is associated with a higher BMI and adiposity [9,10,11]. Conversely, reduced caloric intake at dinner may protect against obesity and metabolic syndrome [12]. Prolonged night fasting duration and increased daily eating episodes were also found to be associated with reduced blood glucose levels [10] and lowered BMI [10,13], respectively. These findings highlight the importance of circadian eating patterns in weight and metabolic regulations. At present, it is unclear whether such eating patterns in pregnancy are associated with postpartum weight outcomes and to our knowledge, previously unexplored.

There has been growing interest in studying the overall diet quality of pregnant women using index scores such as the Healthy Eating Index (HEI) [14], the Dietary Approaches to Stop Hypertension score [15], the New Nordic diet score [16], and the Mediterranean diet score [17]. This approach allows examination of the combined influence of diet on an outcome of interest rather than the influence of single dietary component, which is congruent with the recommendations by the 2015 Dietary Guidelines Advisory Committee [18]. In general, these dietary indices capture the degree of adherence to specific dietary guidelines. With regard to maternal health, most studies have applied this method for evaluating pregnancy outcomes [15,16,17]. Limited studies have explored the use of dietary index in relation to postpartum weight, with one study showing that a better diet quality in pregnancy, ascertained using the HEI, was associated with less postpartum weight retention [14].

Therefore, in the present analysis, we used data from a prospective cohort to analyze circadian eating patterns of pregnant women by assessing their predominant eating period (day vs. night eating), duration of night fasting, and eating episodes per day. We also evaluated the diet quality of women based on the adherence to local dietary guidelines for pregnancy, as measured using the HEI for pregnant women in Singapore (HEI-SGP) [19]. All measures were tested for their associations with substantial PPWR at 18 months.

## 2. Materials and Methods

### 2.1. Study Design and Participants

Data were drawn from the Growing Up in Singapore towards Healthy Outcomes (GUSTO) study (clinicaltrials.gov, NCT01174875) [20]. This study was conducted according to the Helsinki Declaration. Ethics approval was obtained from the Singapore National Health Care Group Domain Specific Review Board (reference D/09/021) and the SingHealth Centralised Institutional Review Board (reference 2009/280/D).

Pregnant women attending antenatal care (≤14 weeks of gestation) from June 2009 to September 2010 at the KK Women’s and Children’s Hospital and National University Hospital in Singapore were recruited. These women were aged ≥18 years and had homogeneous parental ethnic groups (Chinese, Malay or Indian). Women who became pregnant again before 18 months postpartum were excluded from this study. Informed written consent was obtained from all women.

### 2.2. Dietary Assessments

Trained clinic staff conducted the 24-h dietary recall using a five-stage, multiple-pass interviewing technique [21] at 26–28 weeks’ gestation, a timepoint before administrating the routine oral glucose tolerance test (OGTT) to coincide with the participant’s antenatal appointment. Standardized household measuring utensils and food pictures of various portion sizes were provided to assist women in quantifying their food and drink intakes. The time when each meal was consumed was recorded. Total energy intake was assessed using nutrient analysis software (Dietplan, version 7, Forestfield Software, Horsham, UK) containing a local food composition database, with modifications made for inaccuracies found.

In a small subset of participants (*n* = 121), maternal dietary data, which was assessed using a three-day food diary, was also available. We previously showed that circadian eating patterns and diet quality variables derived from the 24-h recall were valid in comparison with the three-day food diary data [22,23]. To retain consistency with analysis methods used in our previous publications [23,24,25] and to increase statistical power, this study presents results based on the 24-h recall data.

#### 2.2.1. Assessment of Circadian Eating Patterns

Circadian eating patterns were assessed in terms of day–night eating, night fasting duration, and eating episodes per day, as described in previous studies [23,24]. Briefly, day- and night-time periods were determined by the local time of sunrise and sunset, occurring at ~0700 h and ~1900 h, respectively, throughout the year in equatorial Singapore (1.352° N). Women were categorized as (i) day eating if they consumed >50% of total energy intake from 07:00–18:59; or (ii) night eating if they consumed >50% of total energy intake from 19:00–06:59 [24]. Night fasting duration was determined based on the longest fasting interval between the consumption of a calorie-containing food or beverage from 19:00–06:59 [23]. The night fasting duration, between 5 h and 12 h, was categorized into three groups as <9 h, 9–10 h, and 11–12 h [23]. Eating episodes were defined as events that provided ≥210 kJ (~50 kcal), with time intervals between eating episodes of ≥15 min [26]. Eating episodes, between 1 and 10, were categorized into four groups as ≤3, 4, 5, and ≥6 based on its distribution.

#### 2.2.2. Assessment of Diet Quality

Diet quality was ascertained using the HEI-SGP. The development and evaluation process was described previously by Han et al. [22] The HEI-SGP is a dietary index that measures adherence to the Singapore dietary guidelines for pregnant women [19]. It consists of 11 components in total—eight assessing adequacy and quality (total fruit, whole fruit, total vegetables, dark green leafy vegetables, orange vegetables, total rice and alternatives, whole grains, dairy, and total protein foods), two assessing nutrients to be taken in moderation (total fat and saturated fat) and a maternal-specific component measuring adherence to antenatal supplements. The total score ranged from 0 to 90, converted to a scale of 0–100. A higher HEI-SGP score reflects greater adherence to dietary guidelines, suggesting a higher diet quality. In our study, women were classified into two groups (lower and higher diet quality) according to the median score at 53.3.

### 2.3. Postpartum Weight Retention

Self-reported pre-pregnancy weight and measured weight at the first antenatal visit (≤14 week’s gestation) were collected. Since maternal measured weight at the first antenatal visit was strongly correlated with reported pre-pregnancy weight (*r* = 0.96; *p* < 0.001) and not subject to recall bias, it was used for analysis in this study [27]. Maternal weights at ≤14 weeks’ gestation and 18 months postpartum were measured using calibrated electronic weighing scales (Seca, Hamburg, Germany) to the nearest 0.1 kg. PPWR was calculated by subtracting the measured weight at ≤14 weeks’ gestation from measured weight at 18 months postpartum [27]. Women were categorized into non-substantial PPWR (<5 kg) and substantial PPWR (≥5 kg) [2].

### 2.4. Covariates

At recruitment and 26–28 weeks’ gestation, trained staff interviewed women on sociodemographic and lifestyle characteristics (e.g., date of birth, ethnicity, education, employment, and number of previous births). Duration and frequency of physical activity were used to derive metabolic equivalent (MET-min/week) scores [28]. Sleep duration and bedtime at night were recorded using the Pittsburgh Sleep Quality Index questionnaire [29] and only administered among a subsample (*n* = 419). Participants self-administered the Edinburgh Postnatal Depression Scale (EPDS) questionnaire to assess mood [30]. Height was measured using a stadiometer (Seca 213, Hamburg, Germany). Early pregnancy BMI was computed from weight at ≤14 weeks’ gestation (kg)/height (m^2^). Serial measurements of maternal weight throughout pregnancy were retrieved from medical notes to estimate gestational weight gain (GWG). An OGTT was used to diagnose gestational diabetes mellitus (GDM) based on the World Health Organization 1999 criteria [31]. The mode of feeding was obtained at three weeks, three months, and six months postpartum, according to the World Health Organization definition [32].

### 2.5. Statistical Analysis

Statistical analyses were performed using IBM SPSS statistics, version 19, or Stata-Corp Stata Statistical Software, release 13. Maternal characteristics were compared between non-substantial and substantial PPWR by Fisher’s exact test for categorical variables and independent *t*-test for continuous variables. Day–night eating, night fasting duration (<9 h, 9–10 h, and 11–12 h), eating episodes (≤3, 4, 5, and ≥6), and diet quality (higher and lower) were simultaneously included in a single model to test for associations with substantial PPWR. Day eating, night fasting for <9 h, ≤3 eating episodes, and higher diet quality were used as reference groups. We estimated odds ratio (OR) and 95% confidence intervals (CI) using the multivariable binary logistic regression.

Confounders were determined from literature review [1,2,3] and identified using directed acyclic graphs. Multivariable models were adjusted for age (years, continuous), BMI (kg/m^2^, continuous), ethnicity (Chinese, Malay, Indian), education (none/primary/secondary, post-secondary, tertiary), parity (0, ≥1), night shift (no, yes), total early pregnancy BMI score (continuous), and total energy intake (kJ, continuous). We did not include employment status (unemployed, employed) or physical activity (<600, 600 to <3000, ≥3000 MET-min/week) in the analyses, as both of these variables were not associated with eating patterns and PPWR.

We further controlled for variables which may be on the pathways between eating patterns and substantial PPWR to examine for any potential mediating effect. These included bedtime (24-h clock, continuous), GDM (no, yes), GWG (kg/week, continuous), and mode of feeding in the first six months postpartum (breastfeeding, mixed feeding, formula feeding). We controlled for bedtime but not sleep duration, as it was not associated with any eating pattern or with substantial PPWR. GWG was estimated using the best linear unbiased prediction method from linear mixed-effects models [33]. The linear trajectory of GWG per week was computed for each individual between 15 and 35 weeks’ gestation [27]. We checked for multicollinearity of all independent variables in the models by screening the correlation coefficients and variance inflation factor (VIF) values. All were below +/−0.40 (Appendix A) and below 2.0, respectively.

Based on the results obtained from the main analysis, we presented the findings focusing on the additive effects of two eating patterns (day–night eating and diet quality) on substantial PPWR. Women were classified into four groups, which were mutually exclusive: (i) Day eating and higher diet quality (*n* = 312); (ii) day eating and lower diet quality (*n* = 284); (iii) night eating and higher diet quality (*n* = 32); and (iv) night eating and lower diet quality (*n* = 59). Day eating and higher diet quality served as the reference group.

Missing values for maternal education (*n* = 8), physical activity (*n* = 5), BMI (*n* = 6), GWG (*n* = 1), total EPDS score (*n* = 17), sleep duration (*n* = 268), bedtime (*n* = 269), GDM (*n* = 23), and mode of feeding (*n* = 13) were imputed 40 times using multiple imputation analyses by chained equations [34]. The number of imputations was determined based on the percentage of missing values. The results of the 40 analyses were pooled using Rubin’s rule [35]. A sensitivity analysis was performed by including only women with a complete dataset for confounders (*n* = 653).

## 3. Results

### 3.1. Maternal Characteristics

Of 1152 enrolled women who conceived natural singleton pregnancies, 126 became pregnant again before 18 months postpartum. Thus, 1026 women were eligible for this study. Of these, 955 women had 24-h dietary recall data, but 11 women with a reported implausible total energy intake of <500 kcal/day (*n* = 3) or >3500 kcal/day (*n* = 8) were excluded, as in previous studies [19,20]. We further excluded 20 women with missing data on early pregnancy weight. At 18 months, 237 women had withdrawn from the study or missed their 18-month visit. A final sample of 687 (67.0%) women was included in this study (Figure 1). Compared with excluded women (*n* = 339; 33%), those included were older (31.3 vs. 29.1 years) and more likely to be parous (64.2% vs. 51.0%) (Appendix A).

Characteristics of women by PPWR status are presented in Table 1. In comparison to women with non-substantial PPWR, those with substantial PPWR were younger (29.5 vs. 31.6 years), more likely to be ethnically Malay (36.4% vs. 22.9%) or Indian (24.5% vs. 16.6%), nulliparous (60.0% vs. 31.2%), had higher early pregnancy BMI (mean 24.8 vs. 23.4 kg/m^2^), greater GWG (mean 0.52 vs. 0.46 kg/week), and were less likely to be diagnosed with GDM (5.5% vs. 21.5%).

Eating patterns and total energy intake of pregnant women by PPWR status are shown in Table 2. In comparison to women with non-substantial PPWR, those with substantial PPWR had a lower HEI-SGP score (mean 49.4 vs. 53.6), reflecting a lower diet quality in pregnancy (62.7% vs. 47.5%). We did not observe significant differences in day–night eating, night fasting intervals, eating episodes, or total energy intake between women of substantial and non-substantial PPWR. Women with a lower diet quality were more likely to practice night eating (17.2% vs. 9.3%; *p* = 0.002), to have 1–3 eating episodes per day (40.2% vs. 22.4%; *p* < 0.001), and to have higher total energy intake (8078 vs. 7703 kJ/day; *p* = 0.037), compared to women with a higher diet quality. No association was observed between diet quality and night fasting intervals (*p* = 0.462).

### 3.2. Associations Between Maternal Eating Patterns and Substantial PPWR

As shown in Table 3 (Model 1), after baseline characteristics adjustment, night eating (OR 1.95; 95% CI 1.05, 3.62) and lower diet quality (1.91; 1.17, 3.10) were independently associated with higher odds of substantial PPWR. Similar findings were observed based on a complete-case analysis (Appendix A). The further inclusion of bedtime (Model 2) attenuated the association between night eating and substantial PPWR (1.82; 0.96, 3.43). The associations remained stable and significant, even after additional adjustment for GDM (Model 3), GWG (Model 4), and mode of feeding (Model 5). Night fasting duration and the number of eating episodes consistently showed no associations with substantial PPWR across models.

The additive effects of day–night eating and diet quality are presented in Figure 2. Women with both night eating and lower diet quality had >3-fold-higher odds of substantial PPWR (3.41; 1.51, 7.69). The complete-case analysis yielded similar findings (3.43; 1.45, 8.11) (Appendix A). Women who practiced night eating with higher diet quality had 2.70-higher odds (0.96, 7.58), while those who practiced day eating with lower diet quality had 2.05-higher odds (1.21, 3.45) of substantial PPWR.

## 4. Discussion

This prospective study examined how circadian eating patterns and diet quality during pregnancy were associated with substantial PPWR, as defined by ≥5 kg at 18 months postpartum among multiethnic Asian women in Singapore. We observed that night eating and lower diet quality in the late second trimester were independently associated with higher odds of substantial PPWR, after adjusting for total energy intake and sociodemographic and lifestyle factors. These findings suggest that besides diet quality, alignment of eating time with day–night cycles during pregnancy may contribute to weight outcomes postpartum.

Compared to examining individual foods or nutrients [36,37], studying diet quality provides a comprehensive picture of the overall dietary patterns which could account for interactions between individual dietary components. Our finding of an association between lower diet quality and substantial PPWR is consistent with a Norwegian cohort, showing that lower adherence to dietary guidelines in pregnancy is associated with increased weight retention at six months postpartum [14]. These data imply that practicing good diet quality through adherence to dietary guidelines may help not only to ensure an adequate nutrient supply for the mother and the fetus, but also to prevent undesirable PPWR.

This study provides new evidence showing that higher calorie intake at night may result in excess PPWR, as supported by review articles [7,38]. Although night eating and lower diet quality were independently linked with weight gain, practicing night eating along with low diet quality demonstrated the greatest odds of substantial PPWR (OR 3.41). Meanwhile, a stronger likelihood of PPWR was observed when night eating was practiced together with higher diet quality (OR 2.70), whereas those practicing day eating with lower diet quality showed a weaker association with PPWR (OR 2.05). However, this finding needs replication and confirmation due to the modest number of women within the group of night eating with higher diet quality (*n* = 32). Nevertheless, it suggests that night eating may be potentially more detrimental than lower diet quality in contributing to substantial PPWR.

Although rodent models have shown that longer fasting time has favorable effects on metabolic health when aligned with sleep-wake cycles [39], we observed no association between night fasting duration and substantial PPWR. Similar to the National Health and Nutrition Examination Survey, prolonged night fasting was not related to BMI despite being linked with reduced levels of diabetes biomarkers [10]. Although several epidemiological studies have shown an inverse relationship between eating episodes and body weight [10,13], we found no association between eating episodes and substantial PPWR. These inconsistent results and paucity of data in pregnant and postpartum women indicate more research is required to clarify the role of night fasting duration and eating episodes on weight management.

There are biologically plausible reasons for the association between night eating and substantial PPWR. High caloric intake at night may alter physiological hormonal secretions (e.g., leptin, ghrelin) and disturb peripheral circadian clocks in various organs (e.g., liver, stomach, adipose tissue), resulting in dysregulation of energy metabolism [40]. The consumption of more calories at night is closely linked with a later bedtime and is associated with overweight and obesity [9,41]. In this study, night-time eaters had a later sleeping time than daytime eaters (2349 h vs. 2306 h, *p* < 0.001), and thus might be more exposed to artificial light at night, leading to delayed or suppressed nocturnal secretion of melatonin, which has been linked with obesity [42]. A reduced thermic effect of food in response to food consumed at night [43] may also contribute to a positive energy balance and weight gain.

Our study has a number of limitations. First, the present findings may not be applicable to other ethnicities and the general population because this study was restricted to Asian women in Singapore and included cohort study volunteers who maybe generally more health conscious. Moreover, differences in characteristics (i.e., age and parity) were noted between included and excluded women, which could raise selection bias and affect the generalizability of findings to a wider population. However, we controlled for these variables in the analysis. Second, dietary data were derived from a single-day 24-h recall, which would limit the ability to assess habitual intake. Although we previously validated the 24-h recall against a three-day food diary for eating patterns in a subsample of women [22,23], we recognized that using multiple dietary recalls or food diaries with newer technologies (e.g., food image recognition apps) across distinct periods of pregnancy would more accurately capture habitual intake while minimizing the respondents’ burden [44] and will consider these in future studies. Third, we did not ascertain postpartum dietary intake, which could differ from the pregnancy period, although some studies have reported that eating habits in pregnancy were likely to be maintained and continued into the postpartum period [45,46]. Finally, the observed relationships could be partly affected by unmeasured or residual confounders, such as light exposure.

## 5. Conclusions

In conclusion, maternal night eating and lower diet quality during pregnancy are associated with substantial PPWR at 18 months. The study suggests that in addition to nutritional quality and quantity of food, the circadian eating pattern may play a role in postpartum overweight and obesity development. There is a possibility that these eating patterns may persist beyond pregnancy and have implications for long-term obesity risk. Hence, further research, specifically larger-scale prospective observational and intervention studies with consideration of postpartum changes in diet and/or experimental studies to explore potential mechanistic pathways, are required. Aligning eating time with day–night cycles and adherence to dietary guidelines during pregnancy may help to alleviate obesity risk in postpartum life.

## Figures and Tables

**Figure 1 nutrients-11-02686-f001:**
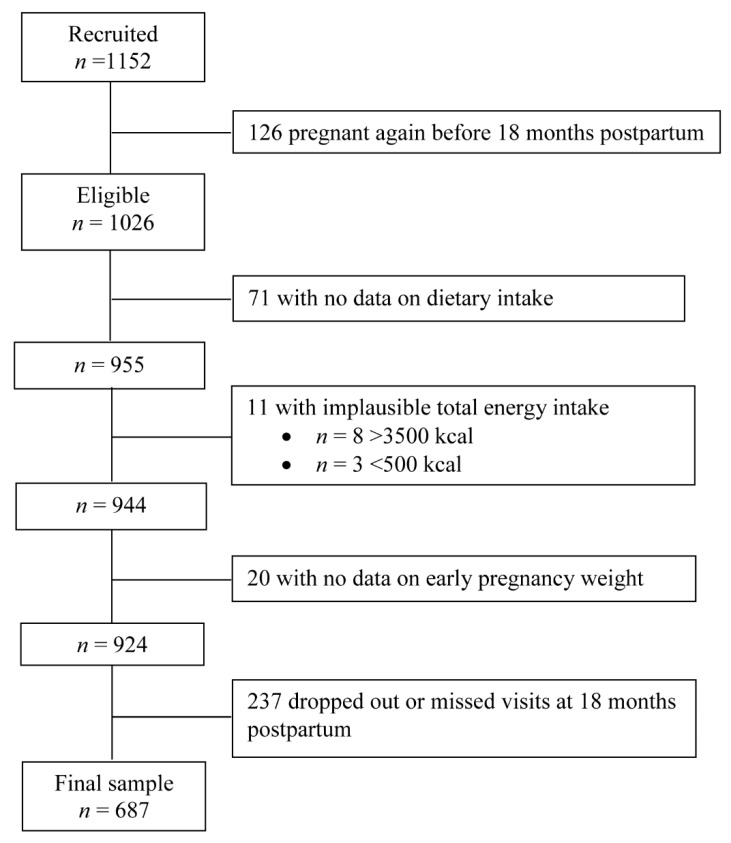
Flowchart of women included for analysis in the Growing Up in Singapore Towards healthy Outcomes (GUSTO) study, Singapore.

**Figure 2 nutrients-11-02686-f002:**
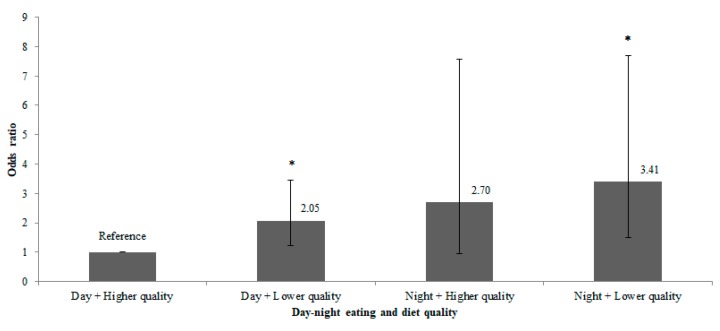
Bar chart presenting the additive effects of day–night eating and diet quality; OR presented are relative to eating pattern of day eating and higher diet quality, OR 1.00). Error bars represent 95% confidence intervals of OR. * *p* < 0.05. Data were analyzed using the multivariable binary logistic regression model, adjusting for maternal age, ethnicity, education, parity, night shift, total Edinburgh Postnatal Depression Scale score, total energy intake, body mass index at ≤14 weeks’ gestation, night fasting intervals, and eating episodes per day.

**Table 1 nutrients-11-02686-t001:** Descriptive characteristics of women from the GUSTO study.

Variable	Total (*n* = 687)	Non-Substantial PPWR <5 kg (*n* = 577)	Substantial PPWR ≥5 kg (*n* = 110)	*p* ^a^
PPWR, kg	1.1 ± 4.1	−0.2 ± 2.9	7.7 ± 2.9	<0.001
Maternal age, years	31.3 ± 5.2	31.6 ± 5.1	29.5 ± 5.1	<0.001
Ethnicity, *n* (%)				<0.001
Chinese	392 (57.1)	349 (60.5)	43 (39.1)	
Malay	172 (25.0)	132 (22.9)	40 (36.4)	
Indian	123 (17.9)	96 (16.6)	27 (24.5)	
Education, *n* (%)				0.433
None/Primary/Secondary	243 (35.4)	199 (34.5)	44 (40.0)	
Post-secondary	216 (31.4)	186 (32.2)	30 (27.3)	
Tertiary	228 (33.2)	192 (33.3)	36 (32.7)	
Parity, *n* (%)				<0.001
0	246 (35.8)	180 (31.2)	66 (60.0)	
≥1	441 (64.2)	397 (68.8)	44 (40.0)	
Employment status, *n* (%)				0.825
Unemployed	227 (33.0)	192 (33.3)	35 (31.8)	
Employed	460 (67.0)	385 (66.7)	75 (68.2)	
Night shift, *n* (%)				>0.950
No	656 (95.5)	551 (95.5)	105 (95.5)	
Yes	31 (4.5)	26 (4.5)	5 (4.5)	
Physical activity, *n* (%)				0.837
<600 MET-min/week	223 (32.5)	187 (32.4)	36 (32.7)	
600 to <3000 MET-min/week	343 (49.9)	286 (49.6)	57 (51.8)	
≥3000 MET-min/week	121 (17.6)	104 (18.0)	17 (15.5)	
BMI at ≤14 weeks’ gestation, kg/m^2^	23.6 ± 4.5	23.4 ± 4.3	24.8 ± 4.9	0.002
Gestational weight gain, kg/week	0.47 ± 0.13	0.46 ± 0.14	0.52 ± 0.13	<0.001
Total EPDS score	7.4 ± 4.5	7.4 ± 4.4	7.3 ± 4.8	0.815
Sleep duration, hours	7.2 ± 1.8	7.1 ± 1.8	7.3 ± 1.8	0.494
Bedtime, 24-h clock	2312 ± 0105	2308 ± 0131	2331 ± 0341	0.120
Gestational diabetes, *n* (%)				<0.001
No	557 (81.1)	453 (78.5)	104 (94.5)	
Yes	130 (18.9)	124 (21.5)	6 (5.5)	
Mode of feeding, *n* (%)				0.454
Breastfeeding	62 (9.0)	53 (9.2)	9 (8.2)	
Mixed feeding	456 (66.4)	387 (67.1)	69 (62.7)	
Formula feeding	169 (24.6)	137 (23.7)	32 (29.1)	

Values are means ± SDs or *n* (%). GUSTO, Growing Up in Singapore Towards healthy Outcomes; PPWR, postpartum weight retention; BMI, body mass index; EPDS, Edinburgh Postnatal Depression Scale. ^a^ Based on independent *t*-test for continuous variables or Fisher’s exact test for categorical variables.

**Table 2 nutrients-11-02686-t002:** Eating patterns and total energy intake of pregnant women at 26–28 weeks’ gestation from the GUSTO study.

Variable	Total (*n* = 687)	Non-Substantial PPWR <5 kg (*n* = 577)	Substantial PPWR ≥5 kg (*n* = 110)	*p* ^a^
Day–night eating, *n* (%)				0.170
Day eating	596 (86.8)	505 (87.5)	91 (82.7)	
Night eating	91 (13.2)	72 (12.5)	19 (17.3)	
Night fasting intervals (hours), *n* (%)				0.889
<9 h	184 (26.8)	154 (26.7)	30 (27.3)	
9–10 h	264 (38.4)	224 (38.8)	40 (36.4)	
11–12 h	239 (34.8)	199 (34.5)	40 (36.4)	
Eating episodes per day, *n* (%)				0.794
1–3 times	215 (31.3)	180 (31.2)	35 (31.8)	
4 times	218 (31.7)	185 (32.1)	33 (30.0)	
5 times	143 (20.8)	122 (21.1)	21 (19.1)	
6–10 times	111 (16.2)	90 (15.6)	21 (19.1)	
HEI-SGP score	53.0 ± 13.9	53.6 ± 14.0	49.4 ± 12.7	0.003
Diet Quality, *n* (%)				0.004
Higher	344 (50.1)	303 (52.5)	41 (37.3)	
Lower	343 (49.9)	274 (47.5)	69 (62.7)	
Total energy intake, kJ/day (1 kcal = 4.186 kJ)	7890 ± 2358	7886 ± 2320	7914 ± 2561	0.908

Values are means ± SDs or *n* (%). GUSTO, Growing Up in Singapore Towards healthy Outcomes; PPWR, postpartum weight retention; HEI-SGP, Healthy Eating Index for pregnant women in Singapore. ^a^ Based on independent *t*-test for continuous variables or Fisher’s exact test for categorical variables.

**Table 3 nutrients-11-02686-t003:** Maternal eating patterns during pregnancy and substantial postpartum weight retention (≥5 kg) at 18 months (*n* = 687).

	Substantial PPWR ≥5 kg				
Eating Patterns	Model 1	Model 2	Model 3	Model 4	Model 5
	OR (95% CI)	OR (95% CI)	OR (95% CI)	OR (95% CI)	OR (95% CI)
Day-night eating					
Day eating	1.00	1.00	1.00	1.00	1.00
Night eating	1.95 (1.05, 3.62)	1.82 (0.96, 3.43)	2.04 (1.07, 3.91)	2.02 (1.05, 3.89)	2.04 (1.06, 3.94)
Night fasting intervals					
<9 h	1.00	1.00	1.00	1.00	1.00
9–10 h	0.81 (0.46, 1.43)	0.88 (0.49, 1.58)	0.87 (0.48, 1.58)	0.85 (0.46, 1.54)	0.87 (0.47, 1.59)
11–12 h	1.19 (0.65, 2.20)	1.32 (0.70, 2.50)	1.34 (0.71, 2.54)	1.26 (0.66, 2.40)	1.29 (0.67, 2.48)
Eating episodes per day					
≤3 times	1.00	1.00	1.00	1.00	1.00
4 times	1.12 (0.61, 2.05)	1.16 (0.63, 2.14)	1.14 (0.61, 2.11)	1.17 (0.63, 2.20)	1.21 (0.64, 2.27)
5 times	1.22 (0.59, 2.50)	1.27 (0.61, 2.62)	1.30 (0.62, 2.71)	1.25 (0.60, 2.62)	1.97 (0.61, 2.71)
≥6 times	1.93 (0.85, 4.37)	1.96 (0.86, 4.49)	2.14 (0.92, 4.97)	2.15 (0.92, 5.03)	2.20 (0.94, 5.17)
Diet quality					
Higher	1.00	1.00	1.00	1.00	1.00
Lower	1.91 (1.17, 3.10)	1.94 (1.19, 3.17)	1.81 (1.10, 2.97)	1.84 (1.11, 3.03)	1.82 (1.10, 3.01)

Analysis was performed using the multivariable binary logistic regression model. All eating patterns were included in a single model and mutually adjusted to each other. PPWR, postpartum weight retention; OR, odds ratio; CI, confidence interval. Model 1: Adjusted for maternal age, ethnicity, education, parity, night shift, total Edinburgh Postnatal Depression Scale score, total energy intake, and body mass index at ≤14 weeks’ gestation. Model 2: Adjusted for Model 1 + bedtime. Model 3: Adjusted for Model 2 + gestational diabetes. Model 4: Adjusted for Model 3 + gestational weight gain through 15–35 weeks of gestation. Model 5: Adjusted for Model 4+ mode of feeding in the first six months.

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
