# Peer review of "Associations of Circadian Eating Pattern and Diet Quality with Substantial Postpartum Weight Retention"

_nutrients, 2019, doi:10.3390/nu11112686_

Round 1

Reviewer 1 Report

Chrononutrition is an important area of research and I would like to commend the authors for undertaking this study to address the important issue of correlates of postpartum weight retention.

The limitation that circadian eating pattern was assessed using only a single 24 hour recall during pregnancy. The comparison to a 3 day food diary does not address questions about the validity of the 24 hour recall as a measure of habitual dietary intake during pregnancy.

Another concern with this study is the correlation between predictors in the models. For example, it is reasonable to assume that bed time and night eating are correlated. A table showing the correlations between predictors is helpful as well as formal evaluation of multicollinearity in the models.

Reviewer 2 Report

This is a very well written paper, that looks at diet quality and time of eating during pregnancy and the impact of PPWR. I believe this manuscript would be of interest to the Nutrients readers. However there are a few comments that may improve the manuscript.

Introduction- I recommend mentioning diet quality and the rational behind it before the final paragraph. As written it feels like an after thought. Methods- Line 113, Why were 26-28 weeks chosen as the time period for the 24-hour recall? Results- I would recommend introducing Table 2 in more detail in your first paragraph or making the introducing/overview of the table its own paragraph. It would be nice to have more about diet quality in the manuscript. For example, which categories did individuals score higher/lower on? Was there one category that appears to be more related to night eating? Discussion- line 291- You state that there was a greater likelihood of PPWR when you had night eating with high diet quality, however, the graph shows that significantly greater odds were seen when a participant had  night eating with low diet quality. Can you confirm which one it is?

Author Response

1. Introduction- I recommend mentioning diet quality and the rationale behind it before the final paragraph. As written it feels like an after thought.

Response: We have added a new paragraph before the final paragraph to discuss diet quality, as below.

Line 91-100: There has been growing interest in studying the overall diet quality of pregnant women using index scores such as the Healthy Eating Index (HEI) [14], the Dietary Approaches to Stop Hypertension score [15], the New Nordic diet score [16] and the Mediterranean diet score [17]. This approach allows examination of the combined influence of diet on an outcome of interest rather than the influence of single dietary component, which is congruent with the recommendations by the 2015 Dietary Guidelines Advisory Committee [18]. In general, these dietary indices capture the degree of adherence to specific dietary guidelines. With regards to maternal health, most studies have applied this method for evaluating pregnancy outcomes [15-17]. Limited studies have explored the use of dietary index in relation to postpartum weight, with one study showing that a better diet quality in pregnancy as ascertained using the HEI was associated with less postpartum weight retention [14].

2. Methods- Line 113, Why were 26-28 weeks chosen as the time period for the 24-hour recall?

Response: We have added explanations as below.

Line 120-122: Trained clinic staff conducted the 24-h dietary recall using a 5-stage, multiple-pass interviewing technique [21] at 26-28 weeks’ gestation, a time-point before administrating the routine Oral Glucose Tolerance Test (OGTT) to coincide with the participant’s antenatal appointment.

3. Results- I would recommend introducing Table 2 in more detail in your first paragraph or making the introducing/overview of the table its own paragraph. It would be nice to have more about diet quality in the manuscript. For example, which categories did individuals score higher/lower on? Was there one category that appears to be more related to night eating?

Response: We provided details for Table 2 in a single paragraph. More details about diet quality have been described in separate studies under the GUSTO cohort (Han et al., 2015; Chia et al., 2018), which we have cited in this paper. Thus, we tried to avoid from repeating the same information here. We have added information about the relationships between diet quality and other eating patterns as below.

Han, C.Y. et al. A healthy eating index to measure diet quality in pregnant women in Singapore: a crosssectional study. BMC Nutr 2015, 1, 39.

Chia, A. et al. Adherence to a healthy eating index for pregnant women is associated with lower neonatal adiposity in a multiethnic Asian cohort: the Growing Up in Singapore Towards healthy Outcomes (GUSTO) study. Am J Clin Nutr 2018, 107, 71-9.

Line 241-249: Eating patterns and total energy intake of pregnant women by PPWR status are shown in Table 2. In comparison to women with non-substantial PPWR, those with substantial PPWR had a lower HEI-SGP score (mean 49.4 vs. 53.6), reflecting a lower diet quality in pregnancy (62.7% vs. 47.5%). We did not observe significant differences in day-night eating, night-fasting intervals, eating episodes and total energy intake between women of substantial and non-substantial PPWR. Women with a lower diet quality were more likely to practice night-eating (17.2% vs. 9.3%; p=0.002), to have 1-3 eating episodes per day (40.2% vs. 22.4%; p<0.001) and higher total energy intake (8078 vs. 7703 kJ/day; p=0.037), compared to women with a higher diet quality. No association was observed between diet quality and night-fasting intervals (p=0.462).

4. Discussion- line 291- You state that there was a greater likelihood of PPWR when you had night eating with high diet quality, however, the graph shows that significantly greater odds were seen when a participant had night eating with low diet quality. Can you confirm which one it is?

Response: It is correct that participants with night-eating + low diet quality had the greatest odds of PPWR. However, we intended to compare the groups between night-eating + higher diet quality vs. day-eating + lower diet quality, relative to the reference group, in relation to PPWR. We have revised the paragraph to make this clearer, as below.

Line 307-315: Although night-eating and lower diet quality were independently linked with the weight gain, practicing night-eating along with low diet quality demonstrated the greatest odds of substantial PPWR (OR 3.41). Meanwhile, a stronger likelihood of PPWR was observed when night-eating was practiced together with higher diet quality (OR 2.70); whereas, those practicing day-eating but with lower diet quality showed a weaker association with PPWR (OR 2.05). However, this finding needs replication and confirmation due to the modest number of women within the group of night-eating with higher diet quality (n=32). Nevertheless, it suggests that night-eating may be potentially more detrimental than lower diet quality in contributing to substantial PPWR.

Round 2

Reviewer 1 Report

I have no additional comments or suggestions.